Leaf spot of Hosta ventricosa caused by Fusarium oxysporum in China

Wang Chunxia 1
Zhang Hulei 1
Wang Shenhai 1
Mao Shengfeng 1 2 maosf@126.com
1 College of Forestry, Nanjing Forestry University , Nanjing, Jiangsu Province , China
2 Co-Innovation Centre for Sustainable Forestry in Southern China, Nanjing Forestry University , Nanjing, Jiangsu Province , China
Atif Rana Muhammad
Electronic publication date: 2021 Dec 7
Publication date: 2021
Volume: 9
Electronic Location ID: e12581
Received 2021 Jun 1; Accepted 2021 Nov 10
Copyright: © 2021 Wang et al.
Copyright year: 2021
Copyright holder: Wang et al.
License: This is an open access article distributed under the terms of the Creative Commons Attribution License, which permits unrestricted use, distribution, reproduction and adaptation in any medium and for any purpose provided that it is properly attributed. For attribution, the original author(s), title, publication source (PeerJ) and either DOI or URL of the article must be cited.
License URL: https://creativecommons.org/licenses/by/4.0/

Keywords: Fungi, Hosta ventricosa, Fusarium oxysporum, Multi gene phylogeny, New disease

Funding: Nanjing Forestry University This work was supported by the second batch of ideological and political demonstration courses of Nanjing Forestry University. The funders had no role in study design, data collection and analysis, decision to publish, or preparation of the manuscript.

==============================
Leaf spot of Hosta ventricosa is a new disease in China. This disease seriously affects the ornamental value and greening function of H. ventricosa. Identification of the causal agent can prevent and control leaf spot in H. ventricosa and promote the healthy development of the H. ventricosa industry. Known incidents of leaf spot of H. ventricosa occurred in three places, and samples were collected. After the fungus were isolated, its pathogenicity was tested according to Koch’s postulates. Isolates ZE-1b and ZE-2b were identified as Fusarium oxysporum based on morphological features and multigene phylogenetic analyses of calmodulin (CMDA), RNA polymerase II subunit A (RPB1), RNA polymerase II second largest subunit (RPB2) and translation elongation factor 1-alpha (TEF1). These results provide a theoretical basis for the control of this disease of H. ventricosa.

Introduction

Hosta ventricosa, is a perennial herbaceous plant of the Hosta genus in Liliaceae. It originated in China, South Korea and Japan (Yu et al., 2015) and is widely distributed in China, including Jiangsu, Anhui, Hebei and other places (Liu et al., 2008). In addition to its bright leaves and graceful habit, this species has strong adaptability to the environment and is suitable for planting under trees, in the shade of buildings or other exposed shaded places. It is an excellent ground cover with ornamental value and greening functions (Zhao, Chen & Lv, 2009). In addition, the whole plant, flowers, leaves or roots can be used as a traditional Chinese medicinal material with the ability to dissipate blood stasis and relieve pain (Zeng, Zhao & Li, 2020).

However, H. ventricosa is impacted by several major pathogens, such as Sclerotium rolfsii Sacc. This disease manifests because the H. ventricosa leaves are especially thick, and during the rainy season, the H. ventricosa rhizome is in contact with water for a prolonged time (Zhao, Chen & Lv, 2009). Thus, the epidermis of the affected area becomes brown and necrotic, and finally a white mycelial layer is formed, which leads to cortex decay (Li et al., 2013). In addition, excessive humidity and poor drainage in the rainy season can also favor diseases caused by Colletotrichum, which mainly damages the leaves, petioles and pedicels of H. ventricosa. The plants present round or nearly round discrolloid spots that are gray or grayish brown (Zhao, Chen & Lv, 2009). Leaf spot of H. ventricosa caused by F. oxysporum is a very serious fungal disease (Fisher et al., 2012). F. oxysporum is one of the top ten most important plant pathogenic fungi in the world, with high virulence and a wide distribution area. The pathogen can cause plant drying and wilting. F. oxysporum can be a saprophytic or parasitic fungus. It is widely found in nature, animals and plants, and has been isolated in cold Arctic areas and arid deserts. This strain can cause wilt and decay of roots, stems, leaves, flowers and fruits in over one hundred plant hosts (Maryani et al., 2019).

Materials and Methods

Experimental materials

Leaves of infected H. ventricosa were collected in Nanjing from 2020 to 2021. Materials used in this study included PDA plates, tissue separation tools, 2% CTAB, and chloroform.

Sampling and isolation

To isolate the fungus, H. ventricose leaves showing leaf spots were collected from three places in Nanjing, China (32°4′47″N, 118°48′31″E; 32°4′45″N, 118°48′31″E; 32°4′44″N, 118°48′31″E.), in September 2020. The collected leaves were rinsed under tap water for 15–30 min. After the leaves were dried, both healthy and affected tissues were cut into small pieces, each of which was 2 × 2 mm in size. The pieces were disinfected in 75% ethanol for 30 s and in 2% NaClO for 90 s, then rinsed with sterile water 3 times for 20 s each time (Si et al., 2021), dried on sterile filter paper and inoculated onto PDA. After the appearance of fungal colonies, blocks of tissue were removed from the edges of the colonies for purification. The morphological characteristics, color, size and shape of the purified colonies were observed and described (Chang et al., 2020). Two single-spore cultures were used for further study and were also deposited in the China Forestry Culture Collection Center (CFCC).

Pathogenicity test

The experiments were replicated 3 times, and a total of 30 seedings were used. Healthy H. ventricosa leaves were collected and rinsed with clean water. The leaves surface were disinfected and dried on an aseptic bench. Pathogenic isolates were inoculated on PDA plates and cultured in an incubator at 25 °C for 5 days. To test the pathogenicity of the isolates, H. ventricosa leaves were wounded with a sterile needle and then inoculated with 5 mm plugs cut out from the growing edges of 5-day-old cultures (Feng et al., 2019). Three replicates were used (Yang et al., 2021b). At the same time, isolates were inoculated onto plants in the natural environment in the wild. Leaves mock inoculated without isolates were used as controls, and the incidence of leaf spot was observed after 3 days (Yang et al., 2021a).

Morphological analysis

Pathogenic isolates were inoculated on PDA plates and incubated in an incubator at 25 °C for 1 week to observe and record the morphology, color, surface characteristics and growth status at the edges of the colonies (Zhang, 2014). The morphology, size and presence of spore septations were recorded under a microscope (Murugan et al., 2020).

DNA extraction, amplification, sequencing and phylogenetic analyses

Before DNA extraction, a small portion of mycelia taken from a 7-day-old cultures of the pathogen grown on PDA plates at 25 °C was collected and transferred to 2-ml Eppendorf tubes. Genomic DNA was extracted by the CTAB method (Freeman, Katan & Shabi, 1996). After passing the test, the mycelia were stored at −18 °C (Guo, Hyde & Liew, 2000; Saghai-Maroof et al., 1984).

The extracted DNA was subjected to polymerase chain reaction (PCR) amplification of partial regions of four genes/regions, namely, calmodulin (CMDA), RNA polymerase II subunit A (RPB1), RNA polymerase II second largest subunit (RPB2) and translation elongation factor 1-alpha (TEF1), which were amplified with primers CL1/CL2A, FA/G2R, 5F2/7CR, and EF1/EF2, respectively (Table 1).

Table 1 Primers for PCR and DNA sequencing.

Locus	Primer	PCR amplification procedures	Reference	
Designation sequence (5′ – 3′)*			
TEF1	EF1	ATGGGTAAGGARGACAAGAC	94 °C to 90 s; Cycles of 94 °C 45 s, 55 °C 45 s, 72 °C 1 min; 72 °C for 10 min; Soak 10 °C	O’Donnell et al. (1998)	
EF2	GGARGTACCAGTSATCATG	O’Donnell et al. (1998)	
CAMD	CL1	GARTWCAAGGAGGCCTTCTC	94 °C to 90 s; Cycles of 94 °C 45 s, 55 °C 45 s, 72 °C 1 min; 72 °C for 10 min; Soak 10 °C	Lombard et al. (2019)	
CL2A	TTTTTGCATCATGAGTTGGAC	Lombard et al. (2019)	
RPB1	Fa	CAYAARGARTCYATGATGGGWC	94 °C to 90 s; Cycles of 94 °C 45 s, 58 °C 45 s, 72 °C 2 min; Cycles of 94 °C 45 s, 57 °C 45 s, 72 °C 2 min; Cycles of 94 °C 45s, 56 °C 45 s, 72 °C 2 min; 72 °C for 10 min; Soak 10 °C	O’Donnell et al. (2010)	
G2R	GTCATYTGDGTDGCDGGYTCDCC	O’Donnell et al. (2010)	
RPB2	5F2	GGGGWGAYCAGAAGAAGGC	94 °C to 90 s; Cycles of 94 °C 45 s, 58 °C 45 s, 72 °C 2 min; Cycles of 94 °C 45 s, 57 °C 45 s, 72 °C 2 min; Cycles of 94 °C 45 s, 56 °C 45 s, 72 °C 2 min; 72 °C for 10 min; Soak 10 °C	O’Donnell et al. (2010)	
7CR	CCCATRGCTTGYTTRCCCAT	O’Donnell et al. (2010)	
Note:

* R = A or G; S = C or G; W = A or T; Y = C or T.

The total volume of the PCR mixture was 50 μL (Lombard, Van & Crous, 2019), containing 19 μL double-distilled water, 2 μL genomic DNA, 2 μL of each primer, and 25 μL Taq DNA polymerase mix. After PCR, the products were sent to Shanghai Jieli Biotechnology Co., Ltd. for DNA sequencing. All sequences with primers CL1/CL2A, FA/G2R, 5F2/7CR, and EF1/EF2 of ZE-1b was deposited in GenBank under accession numbers MW890756, MZ146450, MW890757, MZ088053, and ZE-2b was deposited in GenBank under accession numbers MW885175, MZ127817, MZ126726 and MW885176, respectively.

The CAMD, RPB1, RPB2, and TEF1 sequences were compared to sequences in GenBank using BLAST. The sequences were obtained from GenBank for phylogenetic analyses (Table 2). We downloaded sequences for which the comparison results showed higher than 99% similarity. Using Fusarium aywerte as the outgroup. The arrangement of each gene/region was compared with MAFFTver.7.313 (Katoh & Standley, 2013) and manually adjusted with BioEditver.7.0 (Hall, 1999). It was a combination of these four genes/regions. The ModelFinder was used to select the best-fit model (Kalyaanamoorthy et al., 2017). In IQTree ver.1.6.8, the alternative model of GTR + F + I + G4 was adopted, 1,000 iteration guidance methods were used, and the maximum-likelihood ground method (ML) analysis was used to estimate the system relationship (Nguyen et al., 2015). In the GTR + I + G + F model (2 parallel runs, 2 million generations), MRBayesver.3.2.6 was used for Bayesian analysis. Using burn-in, 25% of sampled data were discard (Ronquist et al., 2012). The phylogenetic trees were drawn with FigTree ver. 1.4.4 (http://tree.bio.ed.ac.uk/software/figtree/).

Table 2 Isolates and sequences used in this study.

GenBank accession	
Species	Culture accession	Host/substrate	Origin	CAMD	RPB1	RPB2	TEF1	
F. acacia-mearnsii	NRRL 26755 = CBS 110255 = MRC 5122	Acacia mearnsii	South Africa	–	KM361640	KM361658	AF212449	
F. armeniacum	NRRL 43641	Horse eye	USA	GQ505398	HM347192	GQ505494	GQ505430	
F. asiaticum	NRRL 13818 = CBS110257 = FRC R-5469 = MRC 1963 = NRRL 31547 T	Hordeum vulgare	Japan	-	JX171459	JX171573	AF212451	
F. atrovinosum	NRRL 34013	Human toenail	USA	GQ505378	–	GQ505472	GQ505408	
	NRRL 34015	Human eye	USA	GQ505380	–	GQ505474	GQ505410	
F. aywerte	NRRL 25410 T	Soil	Australia	KU171417	JX171513	JX171626	KU171717	
F. boothii	NRRL 26916 = ATCC 24373 = CBS 316.73 = NRRL 26855 T	Zea mays	South Africa	-	KM361641	KM361659	AF212444	
F. brachygibbosum	NRRL 34033	Human foot	USA	GQ505388	HM347172	GQ505482	GQ505418	
F. cerealis	NRRL 25491 = CBS 589.93	Iris hollandica	Netherlands	–	MG282371	MG282400	AF212465	
F. chlamydosporum	CBS 145.25 = NRRL 26912 NT	Musa sapientum	Honduras	MN120695	MN120715	MN120735	MN120754	
	CBS 615.87 = NRRL 28578	Colocasia esculenta	Cuba	GQ505375	JX171526	GQ505469	GQ505405	
F. coffeatum	CBS 635.76 = BBA 62053 =
NRRL 20841 T	Cynodon lemfuensis	South Africa	MN120696	MN120717	MN120736	MN120755	
	CBS 430.81 = NRRL 28577	Grave stone	Romania	MN120697	–	MN120737	MN120756	
F. culmorum	NRRL 25475 = CBS 417.86 =
FRC R-8504 = IMI 309344	Hordeum vulgare	Denmark	-	JX171515	JX171628	AF212463	
F. graminearum	NRRL 36905	Triticum aestivum	USA	–	KM361646	KM361664	DQ459742	
F. humicola	CBS 124.73 = NRRL 25535 T	Soil	Pakistan	MN120698	MN120718	MN120738	MN120757	
F. lacertarum	NRRL 20423 = ATCC 42771 = CBS 130185 = IMI 300797 T	Lizard skin	India	GQ505505	JX171467	JX171581	GQ505593	
	CBS 127131	Soil	USA	MN120699	MN120720	MN120739	MN120758	
	NRRL 43680	Contact lens fluid	USA	–	–	EF470046	EF453007	
F. langsethiae	NRRL 53409	Hordeum vulgare	Finland	–	–	HQ154455	HM744667	
F. lunulosporum	NRRL 13393 = BBA 62459 =
CBS 636.76 = FRC R-5822 =
IMI 322097T	Citrus paradisi	South Africa	-	KM361637	KM361655	AF212467	
F. microconidium	CBS 119843 = MRC 839	Unknown	Unknown	MN120700	MN120721	–	MN120759	
F. nelsonii	CBS 119876 = FRC R 8670 =
MRC 4570 T	Plant debris	South Africa	MN120701	MN120722	MN120740	MN120760	
F. nodosum	CBS 200.63	Arachis hypogaea	Portugal	MN120703	MN120724	MN120742	MN120762	
	CBS 201.63 T	Arachis hypogaea	Portugal	MN120704	MN120725	MN120743	MN120763	
F. oxysporum	CBS 144143 T	Solanum tuberosum	Germany	MH484771	-	MH484953	MH485044	
	CFCC 55679 = ZE-1b*	Hosta ventricosa	China	MW890756	MZ146450	MW890757	MZ088053	
	CFCC 55680 = ZE-2b*	Hosta ventricosa	China	MW885175	MZ127817	MZ126726	MW885176	
F. peruvianum	CBS 511.75 T	Gossypium sp.	Peru	MN120707	MN120728	MN120746	MN120767	
F. poae	NRRL 66297		–	–	MG282363	MG282392	–	
F. pseudograminearum	NRRL 28062 = CBS 109956 =
FRCR 5291 = MAFF 237835 T	Hordeum vulgare	Australia	-	JX171524	JX171637	AF212468	
F. sibiricum	NRRL 53429	Avena sativa	Russia	–	–	HQ154471	HM744683	
	NRRL 53430 T	Avena sativa	Russia	-	-	HQ154472	HM744684	
F. sporodochiale	CBS 199.63= MUCL 6771	Termitary	Unknow	MN120709	MN120730	MN120748	MN120769	
	CBS 220.61 = ATCC 14167 =
NRRL 20842 T	Soil	South Africa	MN120710	MN120731	MN120749	MN120770	
F. sporotrichioides	CBS 462.94	Glycosmis citrifolia	Austria	MN120711	MN120732	MN120750	MN120771	
FIESC 24	CBS 101138 = BBA 70869	Phaseolus vulgaris	Turkey	MN120712	MN120733	MN120751	MN120772	
Fusarium sp.	NRRL 13338	Soil	Australia	GQ505372	JX171447	JX171561	GQ505402	
Note:

* Isolates in this study. Ex-type cultures are shown in bold.

T and NT are EX-types.

Results

Incidence of disease and symptoms

The incidence of leaf spot of H. ventricosa in three areas of Nanjing was investigated, and the results showed that the incidence of leaf spot of H. ventricosa in the field was 40%. When the H. ventricosa leaves were infected, the edge of leaves will turn green and yellow, and be dull. With the development of disease, the leaf spots extended and gradually turned yellowish brown.

Pathogenicity of fungal isolates

Based on the colony morphology, fifty fungal samples were divided into seven types. More than 50% are classified as ZE-1b/ZE-2b types. According to the colony morphology, fungi were divided into seven kinds namely ZE-a–ZE-g. According to the ITS sequence ZE-a–ZE-g were identified as Fusarium oxysporum (50%), Fusarium ipomoeae (20%), Fusarium equsiti (10%), Colletotrichum spaethianum (9%), Nigrospora spherica (5%), Colletotrichum gloeosporioide (4%), Colletotrichum siamense (2%). All of the seven kinds of isolates were inoculated seedings, replicated 3 times.

Inoculated H. ventricosa showed leaf spot disease consistent with that observed previously. Two isolates (ZE-1b and ZE-2b) were proven pathogenic to H. ventricosa leaves. Lesions appeared on detached leaves 3 days after inoculation using mycelial plugs (Figs. 1G and 1H). In live plants, 1 week after inoculation, the leaves began to show obvious symptoms of infection, turning yellow and withering (Figs. 1D and 1E). In addition, no lesions were observed on leaves from the control plants (Cong, 2017) (Figs. 1C, 1F). The symptoms on detached leaves and live plants after inoculation were the same as those in the field (Figs. 1A and 1B). The reisolated pathogens from inoculated diseased leaves were consistent with those obtained in the first isolation. Therefore, it was determined that ZE-1b and ZE-2b were the main pathogens causing H. ventricosa leaf spot.

Figure 1 Pathogenicity in detached leaves and in live plants.

(A and B) Diseased leaves naturally infected. (C) No symptoms were observed on leaves from control plants 7 days after inoculation with sterile water; (D) symptoms on live leaves 7 days after inoculation with mycelial plugs of ZE-1b; (E) Symptoms on live leaves 7 days after inoculation with mycelial plugs of ZE-2b; (F) no symptoms were observed on detached leaves from control plants 3, 5, 7 and 10 days after inoculation with sterile water; (G) symptoms on detached leaves 3, 5, 7 and 10 days after inoculation with mycelial plugs of ZE-1b; (H) symptoms on detached leaves 3, 5, 7 and 10 days after inoculation with mycelial plugs of ZE-2b; Bars A = 2 cm; B = 5 cm; C–E = 3 cm; F–H = 1 cm.

Morphological characteristics of fungal isolates

Morphological observations of the pathogenic fungi were carried out. Colonies were inoculated on PDA plates and cultured at 25 °C for 4 days, and the colony diameter was 7 cm (Figs. 2I, 2N). The hyphae grew radially, luxuriously and densely, and the aerial hyphae were velvety, white or pink-white (Liu et al., 2020).

Figure 2 The morphology of hyphae and conidia.

(I) The front and reverse colony morphology of ZE-1b; (J) microconidia of ZE-1b; (K) macroconidia of ZE-1b; (L) chlamydospore formation of ZE-1b; (M) conidiophores of ZE-1b; (N) front and reverse colony morphology of ZE-2b; (O) microconidia of ZE-2b; (P) macroconidia of ZE-2b; (Q) chlamydospore formation of ZE-2b; (R) conidiophores of ZE-2b; Bars I, N = 1 cm; J–M, O–R = 10 μm.

Fusarium has three types of conidia for reproduction and survival under adverse environments: microconidia, macroconidia, and chlamydospores. Microconidia were numerous, oval or kidney-shaped, and scattered, with the size of 4.7–8.6 μm × 2.5–4.7μm (Figs. 2J, 2O). Macroconidia were sickle-shaped, generally symmetrical, slightly curved, and tapering toward the ends, with the size of 23–50.6 μm × 3–5 μm (Figs. 2K, 2P). Chlamydospores were readily produced, with smooth and spherical surfaces (Figs. 2L, 2Q). They were solitary, paired or clustered between hyphae (Du, 2017).

Phylogenetic analyses

Sequences of the genes/regions CAMD, RPB1, RPB2, and TEF1 from the two isolates (ZE-1b and ZE-2b) were deposited in GenBank, and the accession numbers are shown in Table 2. The sequences from ZE-1b and ZE-2b showed 100% similarity with F. oxysporum. These results further indicate that isolation, purification, morphological identification and molecular biology can be used in combination for accurate and reliable results (Cong, 2017).

In the ML phylogenetic tree, two isolates (ZE-1b and ZE-2b) were in the same cluster as F. oxysporum with 100% RAxML bootstrap support values (Fig. 3). The phylogenetic tree obtained by Bayesian analysis was consistent with the ML tree. Bayesian analyses showed that the isolates clustered with F. oxysporum with a high Bayesian posterior probability. Two isolates (ZE-1b and ZE-2b) were identified as F. oxysporum based on multigene phylogeny and morphology.

Figure 3 A maximum parsimony phylogeny for Fusarium oxysporum.

Phylogenetic relationship of ZE-1b and ZE-2b with related taxa derived from maxmum-likelihood (ML) analysis using combined CAMD, RPB1, RPB2, and TEF1 sequence alignment of Fusarium spp., With Fusarium aywerte (NRRL 25410) as the outgroup. RAxML bootstrap support values (ML ≥ 50) and Bayesian posterior probability (PP ≥ 0.80) are shown at the nodes (ML/PP). Ex-type strains are marked in bold. Isolates from H. ventricosa marked in red.

Discussion

In this study, a novel leaf spot disease was studied through pathogenicity determination, morphological identification, and molecular biological identification, and the results showed that the pathogen was F. oxysporum. Herein, wilt of H. ventricosa leaves caused by F. oxysporum was reported for the first time in China.

F. oxysporum is a facultative parasitic fungus that can both infect plants and live in soil (Yang et al., 2021b; Cong, 2017). The transmission of the isolate is either vertical transmission through the mother line to the next generation of seeds or horizontal transmission when the fungi in soil or crops infects the host through wounds. The main means of horizontal transmission are as follows: fungal isolates infect and destroy the vascular bundle from the roots (Foley, 1962) and stems of the plant and spread to various parts of the plant. Due to the exposure of stomata and other external tissues of crops as well as plant wounds, spore and mycelial infection via the air can also occur (Headrick, 1991). Most Fusarium enter through natural openings in plants or seeds, such as stomata (Lin et al., 2014).

F. oxysporum is highly destructive and can destroy many plant organs and cause very severe diseases, such as leaf spot, root rot, stem rot, flower rot and grain wilt (Liu et al., 2020). Globally, F. oxysporum has been identified as a wilt pathogen in many host plants, such as bananas (Maryani et al., 2019; Forsyth, Smith & Aitken, 2006), cotton (Xie et al., 2020; Davis et al., 2006; Zhu et al., 2020), cucumbers (Jaber et al., 2020), sesame (Khalifa, 1997), grapes (El-Sayed, El-Sayed & Eman, 2011), basil (Lori, Malbran & Mourelos, 2014; Mamta et al., 2013; Salim, Salman & Jasim, 2017; Basco et al., 2017), lettuce (Guerrero et al., 2020) and pecan (Rolim et al., 2020). Leaves wilt and eventually drop to the ground, leading to a large area of growth decline; at worst, the whole plant winters and dies, which eventually leads to reductions in yield and quality, causing huge economic losses (Yang et al., 2021a; Li et al., 2020; Cong, 2017).

Originally by scientists abroad, Fusarium was considered a crescent-shaped fungus born on the seed coat. Because many other fungi also produce such sickle-like spores and fungal culture techniques have limitations, the classification of Fusarium has long been in a state of confusion. Later, German scientists introduced the first systematic classification of Fusarium and proposed a relatively complete classification system based on the biological characteristics of these fungi, combined with their morphological structures, which laid the foundation for classification research on Fusarium (Du, 2017). Fusarium was initially divided into 44 strains, with 35 strains in China, laying a foundation for the study of Fusarium here (Yu, 1977). Currently, more than 3,000 strains of Fusarium have been studied, 40 physiological strains have been identified and collected, and one new strain was found. Twenty-eight strains of Fusarium zhejiangensis were identified in Zhejiang, and Fusarium zhejiangensis was first recorded in literature. “Fusarium disease in Taiwan” was published in Plant Pathology, Chung Hsing University, Taiwan (Du, 2017).

In recent years, research on Fusarium taxonomy in China has developed rapidly based on both morphology and molecular biology. This experiment provides a basis for field prevention and treatment of H. ventricosa leaf spot caused by Fusarium and provides a reference for further genetic analysis and cultivation of disease resistant varieties of H. ventricosa (Zhi, 2020).

Conclusion

This is the first report of H. ventricosa leaf spot in China and Chinese H. ventricosa is a new host of F. oxysporum. We should take reasonable preventive measures against diseases. This study provided theoretical guidance for the control of Chinese H. ventricosa leaf spot.

Additional Information and Declarations

Competing Interests

Author Contributions

Data Availability

The authors declare that they have no competing interests.

Chunxia Wang performed the experiments, analyzed the data, prepared figures and/or tables, authored or reviewed drafts of the paper, and approved the final draft.

Hulei Zhang performed the experiments, prepared figures and/or tables, and approved the final draft.

Shenhai Wang performed the experiments, prepared figures and/or tables, and approved the final draft.

Shengfeng Mao conceived and designed the experiments, analyzed the data, authored or reviewed drafts of the paper, and approved the final draft.

The following information was supplied regarding data availability:

The data is available at NCBI: MW890756, MZ146450, MW890757, MZ088053, MW885175, MZ127817, MZ126726 and MW885176.

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
