# Peer review of "Leaf spot of Hosta ventricosa caused by Fusarium oxysporum in China"

_PeerJ, doi:10.7717/peerj.12581_

## Round 0.1 · original submission · Major Revisions

Please address the concerns and suggestions raised by the reviewers.

·

Basic reporting

The authors isolated and identified a new pathogen causing leaf spot of Hosta ventricosa in China. The morphological and molecular characters of the isolates indicated that the pathogen causing leaf spot of Hosta ventricosa is Fusarium oxysporum. Additionally, Koch’s postulates were also fulfilled by artificially inoculation. I appreciate the works done in this research.  However, the manuscript will benefit from editing by a native English speaker to polish the language. 
1. For the figure legend of Figure 3, Should it be "Fusarium proliferatum" or "Fusarium oxysporum"

Experimental design

1. For Sampling and isolation methods (Line49), the deatiled informations of three sampling places should be provided (including the longitude and latitude)
2. For Pathogenicity test (Line59), how many plants were used for pathogenicity test should be clarified.
3. For line105, The author isolated fifty fungal samples and divided them into seven types base on the colony morphology, The results and the detailed classification standard should be provided.

Validity of the findings

1. F. oxysporum is important and destructive pathogen with a broad host. This is the first report of H. ventricosa leaf spot in China and Chinese H. ventricosa is a new host of F. oxysporum. Therefore,The incidence and the epidemic features of the leaf spot disease caused by the F. oxysporum should be provided in the results part.

2. ZE-1b and ZE-2b are all Fusarium oxysporum, but the disease symptomes caused by ZE-1b and ZE-2b were not the same (Figure 1). Are there any discrepancies in pathogenicity among the isolates.

Reviewer 2 ·

Basic reporting

The manuscript titled "Leaf spot of Hosta ventricosa 1 caused by Fusarium oxysporum in China" was submitted for publication in the PeerJ. It reports the first occurrence of leaf spot disease caused by this particular pathogen in China.

Experimental design

no comment

Validity of the findings

The findings seem valid; however, a single pathogen first report cannot be published in PeerJ.

Additional comments

The article fails to meet our standards as the authors have just provided a first national report of a disease in an ornamental plant. This kind of work could be sent for publication to Plant pathology journals like "Plant Disease" or "Plant pathology" and some other related journals which publish first disease notes from various countries. I regret to state that this paper could not be published in the PeerJ as it is a short first disease report based on only morphological, molecular, and pathogenicity characterization.

Reviewer 3 ·

Basic reporting

Basic Reporting:
Leaf spot disease of ornamental plant, H. ventricosa caused by fungal pathogen F. oxysporum is reported. For more confidence in results, the experiment was designed to perform on three places. The reporting is based on two important techniques, infection tests under control conditions on host plants and identification of Pathogenic isolate with sequencing by targeting multiple genes.
This is good data for first report. But For a journal having repute of publishing extensive research, the other parameters like passage of pathogen in the plant, lab and field management through modern pathogen approaches are important.

Experimental design

Since, disease was just confirmed through identification of the pathogen on the basis of physical characteristics and infection tests, so up to these two points the experiment designing is good. At the same time, our previous knowledge directs that Fusarium oxysporum is mainly a soil borne fungus and causes rots and wilts more commonly but exceptional diseases leaf spots are interesting.

Validity of the findings

Findings are straightforward and easily reproducible. Usually F. oxyxporum causes rots and wilts. The proposed study is novel regarding leaf spot role of fusarium as less number of such reports are available.

Additional comments

I would suggest accepting the paper after revising the grammar and English language and will also suggest including one or two more genes to target for sequence which will definitely put some weight to publish this manuscript in this prestigious journal.

---

## Round 0.2 · Minor Revisions

Please address the query raised by one of the Reviewers regarding the Figure.

·

Basic reporting

no comment

Experimental design

no comment

Validity of the findings

There is another question for Figure 1G, the authors inoculated two mycelial plugs of ZE-1b on the leaves of H. ventricosa, why does the plug on the left side not infect the leaf.

Additional comments

The authors properly addressed some the comments that I made. However, I think the pathogenicity experiment lacked enough trust. The detailed comment that I feel they need clarification is as follows.
There is another question for Figure 1G, the authors inoculated two mycelial plugs of ZE-1b on the leaves of H. ventricosa, why does the plug on the left side not infect the leaf?

Reviewer 3 ·

Basic reporting

I have read the whole revised manuscript. The new version is improved. Professional English has been incorporated throughout the article. References are updated. Whole story is presented precisely and in a good way. All the figures and tables are mentioned in the text and properly numbered. Raw data regarding accession number of the sequences etc. is provided with valid links.

Experimental design

The research fit well into scope of the journal. Since such reports are rare, so it would be a good addition in the coming issue/s. Research question was raised as hypothesis and then answered with valid results. The research addresses the important problem of Leaf spot disease which was noticed by authors and his team during survey and was later monitored for isolation of the pathogen and its valid identification. It was proved that the disease caused by Fusarium oxysporum in the Hosta ventricosa through infection tests and through sequencing of multiple genes fusarium was declared as causal organism. The hypothesis was further answered through infection tests on the host plant and microscopy strengthened, which not only validate Fusarium as causal organism but also provide enough information regarding its virulence against H. ventricosa. The data provided was replicated and the method could be reproduced easily in any lab having enough facilities.

Validity of the findings

According to my knowledge, it is novel work, since it is being reported first time which is enough proof for its novelty and need. As I mentioned earlier, the authors incorporated all the doable suggestions which I humbly suggested during the review of initial draft. The word file clearly shows the technical incorporations and improvement in the write up.

---

## Round 0.3 · accepted · Accept

After the incorporation of said changes in the manuscript, the manuscript can now be published in the journal.